# Cerebellar repetitive transcranial magnetic stimulation has no effect on contraction-induced facilitation of corticospinal excitability

Akiyoshi Matsugi[1]*, Aki Tsuzaki[1], Soichi Jinai[1], Yohei Okada[2], Nobuhiko Mori[3], Koichi Hosomi[3,4]

1 Faculty of Rehabilitation, Shijonawate Gakuen University, Daitou City, Osaka, Japan, 2 Neurorehabilitation Research Center of Kio University, Koryo-cho, Kitakatsuragi-gun, Nara, Japan, 3 Department of Neurosurgery, Osaka University Graduate School of Medicine, Suita City, Osaka, Japan, 4 Department of Neurosurgery, Toyonaka Municipal Hospital, Toyonaka City, Osaka, Japan

* a-matsugi@reha.shijonawate-gakuen.ac.jp

**Data Availability Statement:** The datasets generated and analyzed in the current study are available from Mendeley Data (Matsugi, Akiyoshi (2023), "CIF_CrTMS", Mendeley Data, V1, doi: 10.17632/z27jbr69bs.1).

## Abstract

This study aimed to investigate whether the cerebellum contributes to contraction-induced facilitation (CIF) of contralateral corticospinal excitability. To this end, repetitive cerebellar transcranial magnetic stimulation (TMS) was used to test whether it modulates CIF. Overall, 20 healthy young individuals participated in the study. Single-pulse TMS was applied to the left primary motor cortex to induce motor-evoked potentials (MEP) on electromyography of the right first dorsal interosseous (FDI) muscle to test corticospinal excitability. This measurement was conducted during contraction (10% maximum voluntary contraction [MVC]) and rest (0% MVC) of the FDI muscle. CIF, cerebellar brain inhibition (CBI), cortical silent period (cSP), and resting motor threshold (rMT) were measured before and after low-frequency repetitive TMS (crTMS) of the right cerebellum to downregulate cerebellar output. The CIF (contraction/rest of the MEP), CBI (conditioned/unconditioned MEP) during contraction, cSP, and rMT were not affected by crTMS. At rest, CBI was decreased. These findings indicated that the primary motor cortex function for the increase in corticospinal excitability was not affected by crTMS. This study contributes to our understanding of the role of the cerebellum in motor control. Additionally, it may inform decision-making for the site of cerebellar ataxia treatment using non-invasive brain stimulation.

## Introduction

The cerebellum contributes to voluntary motor control, particularly force control [1]. Force control is impaired in patients with spinocerebellar disease (SCD), demonstrating a close relationship between force control and the cerebellum [2]. However, evidence of the effects of cerebellar degeneration and modulation of cerebellar output excitability in the motor cortex or corticospinal excitability during force control is lacking.

A functionally interactive loop between the cerebellum and contralateral primary motor cortex helps modulate the excitability of intracortical facilitatory and inhibitory neural circuits

**Funding:** JSPS KAKENHI (grant number: 23K10418). . The funders had no role in study design, data collection and analysis, decision to publish, or preparation of the manuscript.

[3]. Therefore, these motor cortex functions are altered in patients with cerebellar damage even when the motor cortex is intact [4, 5]. For example, the resting motor threshold (rMT) estimated by transcranial magnetic stimulation (TMS) is increased in patients with cerebellar damage [6], indicating suppression of excitability in pyramidal tract neurons, spinal motoneuron, and facilitatory neural circuit in the primary motor cortex [7, 8]. Prior to starting a physical movement, corticospinal excitability must be quickly increased right before the movement. However, this mechanism, estimated by the modulation of motor-evoked potential (MEP) induced by TMS, does not successfully occur in patients with spinocerebellar degeneration (SCD) [9]. Furthermore, cerebellar inhibitory output, estimated by cerebellar brain inhibition (CBI) using the pair pulse TMS method [10], is reduced during actual motor execution [11, 12] and imagery of motor execution [13]. Moreover, to regulate the output force [14, 15], GABAergic inhibitory neural circuits in the primary motor cortex, estimated by the cortical silent period (cSP) [16], are abnormally prolonged in these patients [4–6, 17]. These recent (2006–2023) findings are associated with corticospinal excitability, rMT, CBI, and cSP estimated with TMS, strongly supporting the hypothesis that the cerebellum influences contralateral motor cortex function for the modulation of corticospinal excitability for force control. However, the contribution of the cerebellum to the contraction-induced facilitation (CIF) of corticospinal excitability remains unclear.

CIF is a unique method of estimating the capacity of corticospinal tracts to increase excitability by adding TMS to the motor cortex during muscle contraction and observing the amount of facilitation over resting MEPs. Whilst it is a potential method of inferring motor cortex function during muscle contraction, it is more difficult to interpret than resting state MEP, because it involves many inputs from the motor cortex and activity of intracortical excitatory and inhibitory circuits [18]. The lack of increment of MEP size during muscle contraction in people with degenerative neurological disease is interpreted as defective facilitatory cortical inputs [19, 20]. Therefore, failure of increment of MEP size for contraction in people with cerebellar ataxia [9] can be due to a change of output from the cerebellum. To investigate this hypothesis, we can use repetitive TMS (rTMS) as a neuromodulation technique. The involvement of the cerebellum in the regulation of corticospinal tract excitability can be studied by observing the modulation of the CIF by the neuromodulation technique of rTMS to the cerebellum.

rTMS is often used to investigate the contribution of the cerebellum to human motor control. The effects of rTMS vary depending on frequency and pulse pattern. Continuous theta burst stimulation, represented by the high-frequency rTMS protocol, to the lateral cerebellum, induces a reduction of short intracortical inhibition (SICI) and long intracortical inhibition (LICI) [21]. Low-frequency rTMS (1 Hz) of the cerebellum suppresses contralateral short-interval intracortical inhibition [22] and increases intracortical facilitation [23] in the resting state of healthy humans. Therefore, cerebellar rTMS may affect the contralateral intracortical inhibitory and facilitatory circuits for the modulation of corticospinal excitability in the resting state of humans. Furthermore, 1Hz rTMS can inhibit coordinated head and eye movements [24] and motor learning by reaching the wrist [25]. Similarly, 1Hz rTMS of the cerebellum can reveal its impact on behavior and response to stimuli. Based on recent studies, we used 1Hz rTMS to investigate the contribution of the cerebellum to CIF, which reflects the function of increments in corticospinal excitability for active muscle contraction.

Recent reports (2021–2023) suggest that non-invasive brain stimulation (NIBS) techniques [26], such as rTMS [27] and transcranial electrical stimulation [28, 29], can improve the cerebellar ataxia. Several hypotheses have been proposed to explain this phenomenon. One possibility is that NIBS alters cerebellar cortical function and modulates the excitability of cerebellar output. This neuromodulation montage in the cerebellum may affect remote brain regions,

such as the primary motor cortex. The function of the motor cortex plays an important role in movement control and may be one of the reasons for its therapeutic effects in cases of cerebellar damage. Cerebellar hemisphere NIBS may influence the contralateral motor cortex during muscle contraction. However, the mechanism by which cerebellar NIBS influences corticospinal excitability during muscle contraction remains unclear. The finding that NIBS of the unilateral cerebellar hemisphere can modulate contralateral motor cortex function, particularly CIF, rMT, and cSP, could provide evidence for selecting the target site for the treatment of cerebellar ataxia caused by stroke or degenerative disease.

In this study, we aimed to examine whether CIF is affected by 1Hz cerebellar rTMS, which downregulates cerebellar output activity, to verify the hypothesis that the cerebellum contributes to the CIF of corticospinal excitability estimated by MEP. If Cerebellar rTMS no longer causes CIF, then the cerebellum is considered to contribute to CIF. Conversely, if CIF is not affected in any way, then cerebellar involvement in CIF is considered to be minor.

## Materials and methods

### Ethics statements

This study was approved by the Ethics Committee of Shijonawate Gakuen University (No: 22–9). All participants were thoroughly briefed on the experiment and provided written informed consent prior to participation. This study was conducted under the principles and guidelines of the Declaration of Helsinki.

### Participants

Twenty healthy individuals participated in the study. The experiments were conducted, and data were obtained between 27/2/2023 and 28/3/2023. All the participants were right-handed. The participants were randomly assigned to either the Sham-rTMS group (n = 10) or Active-rTMS group (n = 10) for between-participant design to avoid the effect of confounding variable such as use-dependent learning influences in the force control task. Before the experiment, the optimal inter-stimulus interval (ISI) at which an individual would observe CBI was determined. Individuals who did not observe CBI within 4–9 ms were excluded from subsequent experiments; two individuals were excluded based on this criterion.

### Experimental setting

Fig 1A illustrates the experimental setup. Participants were seated in a chair with the backrest facing a computer monitor. Their right hand was secured to a metal frame to prevent unwanted movements after TMS. The abduction force of the right index finger was measured using a force transducer, and the participants were instructed to relax all the other hand and arm muscles. Before the experiment, the maximum voluntary contraction (MVC) force level of the index finger with contraction of the first dorsal interosseous (FDI) muscle and the target force level was set to either 0% (rest condition) or 10% (contraction condition) of the MVC. During the experiment, participants were instructed to adjust their force level to the target displayed on a monitor in front of them.

For the acquisition of electromyography (EMG) signals, two Ag/AgCl surface-recording electrodes were strategically positioned with a separation of 2 cm on the right First Dorsal Interosseous (FDI) muscle. Signal amplification was executed through the utilization of a MEG-1200 amplifier (Nihon Kohden, Tokyo, Japan) equipped with a passband filter spanning 15–3 kHz. The amplified EMG signals underwent analog-to-digital conversion at a sampling rate of 10 kHz using a PowerLab 800S A/D converter (AD Instruments, Colorado Springs,

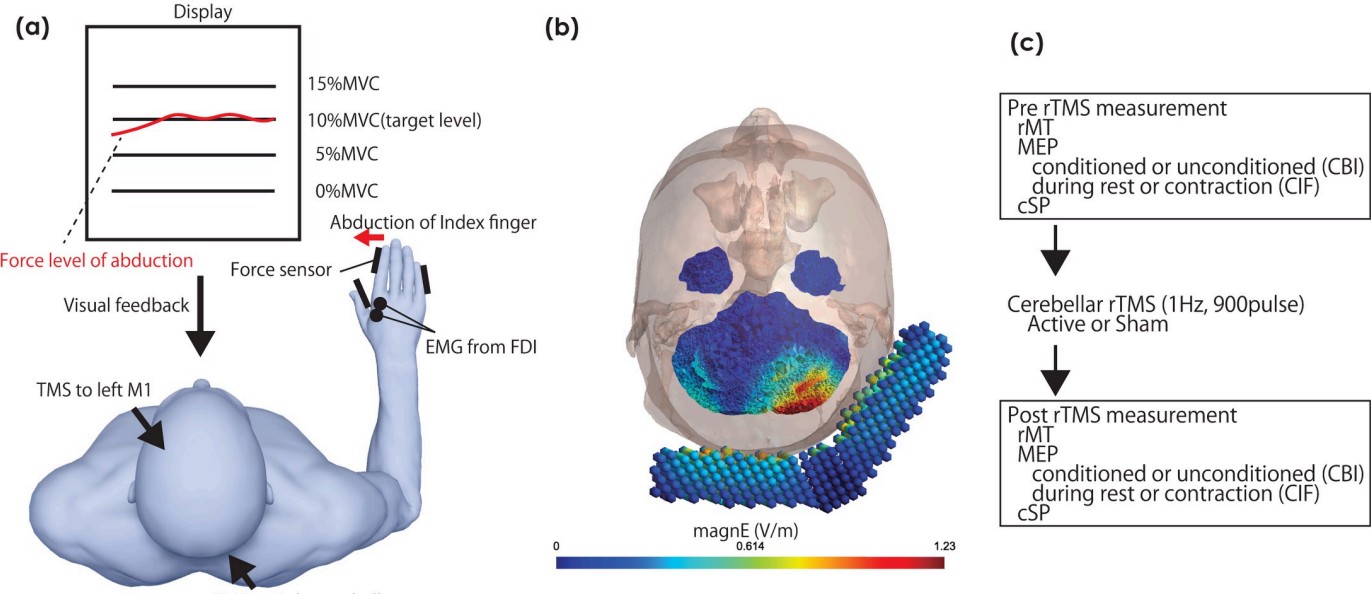

**Fig 1.** Experimental setup (A), simulation of the electrical field by cerebellar TMS (B), and general procedure (C). (A) Experimental setup: EMG, electromyography; FDI, first dorsal interosseous; MVC, maximum voluntary contraction of muscle. (B) Visual Representation of Transcranial Magnetic Stimulation (TMS) Electrical Field in the Cerebellum: Horizontal Perspective. In this illustrative depiction, the double-cone coil, denoted by a composite of small circles in blue, symbolizes the apparatus employed for TMS. The magnitude of the electrical field is graphically represented by the labeled unit MagnE (V/m). The region characterized by a robust electrical field is highlighted in red. This visual simulation provides insight into the spatial distribution and strength of the electrical field generated by the TMS application in the cerebellum when viewed horizontally. (C) General experimental procedure: The resting motor threshold (rMT), cortical silent period (cSP), and motor-evoked potential (MEP) during rest/contraction (to calculate contraction-induced facilitation [CIF]) with/without conditioning cerebellar TMS (to calculate cerebellar brain inhibition [CBI]) are evaluated before and after repetitive transcranial magnetic stimulation over the right cerebellum.

CO, USA) and were subsequently stored digitally on a personal computer. It is worth noting that this methodology for EMG signal recording aligns with the procedures outlined in a previous study conducted by our research group, which ensures methodological consistency and facilitates comparisons across investigations [30].

## Single-pulse TMS over the primary motor cortex (test stimulation)

A TMS pulse was administered to the primary motor cortex using a figure-of-8 coil (D70; Magstim Company Ltd., Spring Gardens, UK) connected to a magnetic stimulator (Magstim; Magstim Company Ltd., Spring Gardens, UK). Precise placement of the figure-of-8 coil centrally over the hotspot within the left primary motor cortex aimed to elicit Motor Evoked Potentials (MEP), specifically targeting the right First Dorsal Interosseous (FDI) muscle. The hotspot, representing the location in the left hemisphere where the participant exhibited the maximal MEP response, was methodically identified on the participant's head. To maintain uniformity and prevent any changes at the stimulation site, a designated hotspot was carefully marked on the participant's swim cap. This procedural precaution ensured the reliability and stability of the stimulation site throughout the experimental sessions. Furthermore, the current flow within the coil was directed in the anterior to posterior direction, resulting in the generation of a posterior-anterior traveling current within the brain. These systematic approaches were implemented to uphold the scientific rigor and integrity of the experimental methodologies [31, 32]. The rMT of the FDI muscle was defined as the minimum intensity of the magnetic stimulator output that produced MEPs with an amplitude >50 μV in at least three of five

stimulations delivered over the hotspot of the left primary motor cortex [14, 33]. Consistent with previous studies [34], the stimulation intensity used in this experiment was set to elicit a test MEP size of 0.5–1 mV to measure CBI before experiment.

### Single-pulse TMS over the cerebellum (conditioning stimulation)

Participants were instructed to maintain a sitting position on a chair. The center of the junction of the double-cone coil (D-B-80; MagVenture, Farum, Denmark) connected to a magnetic stimulator (MagPro R20; MagVenture) was set 1 cm below and 3 cm to the right of the inion to stimulate the right cerebellar hemisphere [10, 35, 36]. Previous studies have shown that an upward current applied to the cerebellum can effectively stimulate this region [10], allowing the upward delivery of the current to the brain [37].

In Fig 1B, the visual representation illustrates the simulated magnitude of electrical fields within the horizontal view of the right cerebellar hemisphere. This simulation was generated by employing a double-cone coil (D-B-80) and simulated through the utilization of SimNIBS software (version 4.0.0) [38, 39] under the default settings. This simulation indicates that TMS using a double-cone coil can selectively stimulate the right cerebellar hemisphere.

TMS intensity for stimulating the cerebellum was set to 90% of the rMT in the right FDI relative to the right cerebellum [13]. The rMT for cerebellar TMS was defined as the lowest stimulation intensity producing a short-latency EMG response and cervicomedullary MEP in the right FDI muscle immediately after cerebellar TMS for five of ten consecutive stimuli [13]. To measure CBI, the optimal ISI at which maximal MEP inhibition was observed was 4–9 ms [34, 40].

### Repetitive TMS over the cerebellum (neuromodulation)

The same magnetic stimulator and coil were used for the CBI. The intensity and site of the coil were set in the same manner as those for the CBI. The ISI was set at 1 s, and 900 pulses were applied [25, 41, 42] because 1-Hz rTMS over the cerebellum immediately reduces motor function [43–45]. For sham rTMS, the coil was held at a 90˚ angle from the scalp over the same location used for active stimulation [25, 45].

### General procedure

Fig 1C shows the experimental procedure. Initially, rMT was measured. Then, MEP was measured during FDI contraction—by maintaining a force of 10% of the MVC level—or at rest (0% MVC level), with or without conditioning with single-pulse cerebellar TMS. For each condition (rest without cerebellar TMS, contraction with cerebellar TMS, rest with cerebellar TMS, and contraction without cerebellar TMS), 10 MEPs were measured. Contractions were measured after resting conditions. Finally, rTMS intervention was performed, and the rMT and MEP were measured.

### Data analysis

MEP amplitude was measured as the peak-to-peak amplitude of the potential approximately 20 ms after TMS of the primary motor cortex. The cSP was measured only during muscle contraction. An EMG burst can be detected after the silent period. The cSP duration was defined as the interval between the primary motor cortex TMS and EMG burst onset [14, 46, 47]. One analyst visually identified them according to the following definition: bursts of EMG showing a cSP offset > mean + 3 × standard deviation of the pre-stimulus background EMG level in the resting state [14, 47]. CBI was calculated using conditioned/unconditioned MEP amplitudes. CIF was calculated using the MEP amplitude of the contraction/rest condition.

To estimate the facilitation of MEP via contraction and inhibition by cerebellar paired TMS for CBI, a one-sample t-test (two-sided) was conducted for CIF and CBI with Bonferroni correction. We assessed the effect of rTMS (Sham or Active) and time (pre or post) on rMT for motor cortex stimulation, rMT for cerebellar stimulation, CIF, and cSP by applying a linear mixed-effect model (Satterthwaite method) fitted with maximum likelihood with a random intercept per participant. If significant interaction were observed, post-hoc comparisons was conducted to estimate the difference between factors. The JASP software (version 0.17.1; University of Amsterdam, Amsterdam, Netherlands) [48] was used to perform all statistical analyses, with the alpha level set at 0.05.

## Results

The mean age of the participants was 20.4±0.5 years, with a 1:1 male-to-female ratio (10 and 10, respectively). No harmful rTMS-related side effects were observed. The ISI for CBI was 6 ±2 ms.

### (a) rMT

Fig 2A and Table 1 show the results of the rMT for motor cortex stimulation. In the linear mixed-effect model, the model fit statistics were: deviance = 254.072, log of likelihood = -127.036, df = 6, Akaike Information Criterion (AIC) = 266.072, and Bayesian Information Criterion (BIC) = 276.205. These results indicate that this model is acceptable for testing the effects of rTMS and time, and excludes the random effects of the grouping factor: "participant." This model indicated no fixed effect of time (estimate = 0.25, standard error [SE] = 0.339, df = 20, t = 0.738, p = 0.469), rTMS (estimate = $7.592^* \times 10^{-14}$, SE = 2.477, df = 20, t = $3.065 \times 10^{-14}$, p = 1) and interaction between time and rTMS (estimate = 0.2, SE = 0.339, df = 20, t = 0.59, p = 0.562).

Table 2 shows the results of rMT for cerebellar stimulation. In the linear mixed-effect model, the model fit statistics were as follows: deviance = 177.013, log of likelihood = -88.507, df = 6, AIC = 189.013, and BIC = 199.147. These results indicate that this model is acceptable for testing the effects of rTMS and time and excludes the random effects of the grouping factor: "participant." This model indicated no fixed effect of time (estimate = -0.125, SE = 0.103, df = 18, t = -1.213, p = 0.241), rTMS (estimate = -0.775, SE = 1.287, df = 18, t = -0.602, p = 0.555), and interaction between time and rTMS (estimate = -0.075, SE = 0.103, df = 18, t = 0.728, p = 0.476).

### (b) CBI

Fig 3 shows typical waveform of MEPs with and without conditioning single-pulse cerebellar TMS, whereas Fig 2B and Tables 3 and 4 show the CBI results. The two-tailed one-sample t-test revealed significant inhibition under pre- and post-sham-rTMS rest conditions, whereas only the pre-rTMS condition exhibited significant inhibition under active-rTMS conditions (Table 3). In the linear mixed-effect model, the model fit statistics were: deviance = 3.263, log of likelihood = -1.632, df = 15, AIC = 33.263, and BIC = 68.994. These results indicate that this model is acceptable for testing the effects of rTMS, time, and contraction conditions and excludes the random effects of the grouping factor: "participant." This model indicated no fixed effects for each factor or interaction (Table 4).

### (c) CIF

Fig 2C and Table 5 present the CIF results. A one-sample t-test revealed significant facilitation under all conditions (Table 5). The model fit statistics were: deviance = 155.233, log of

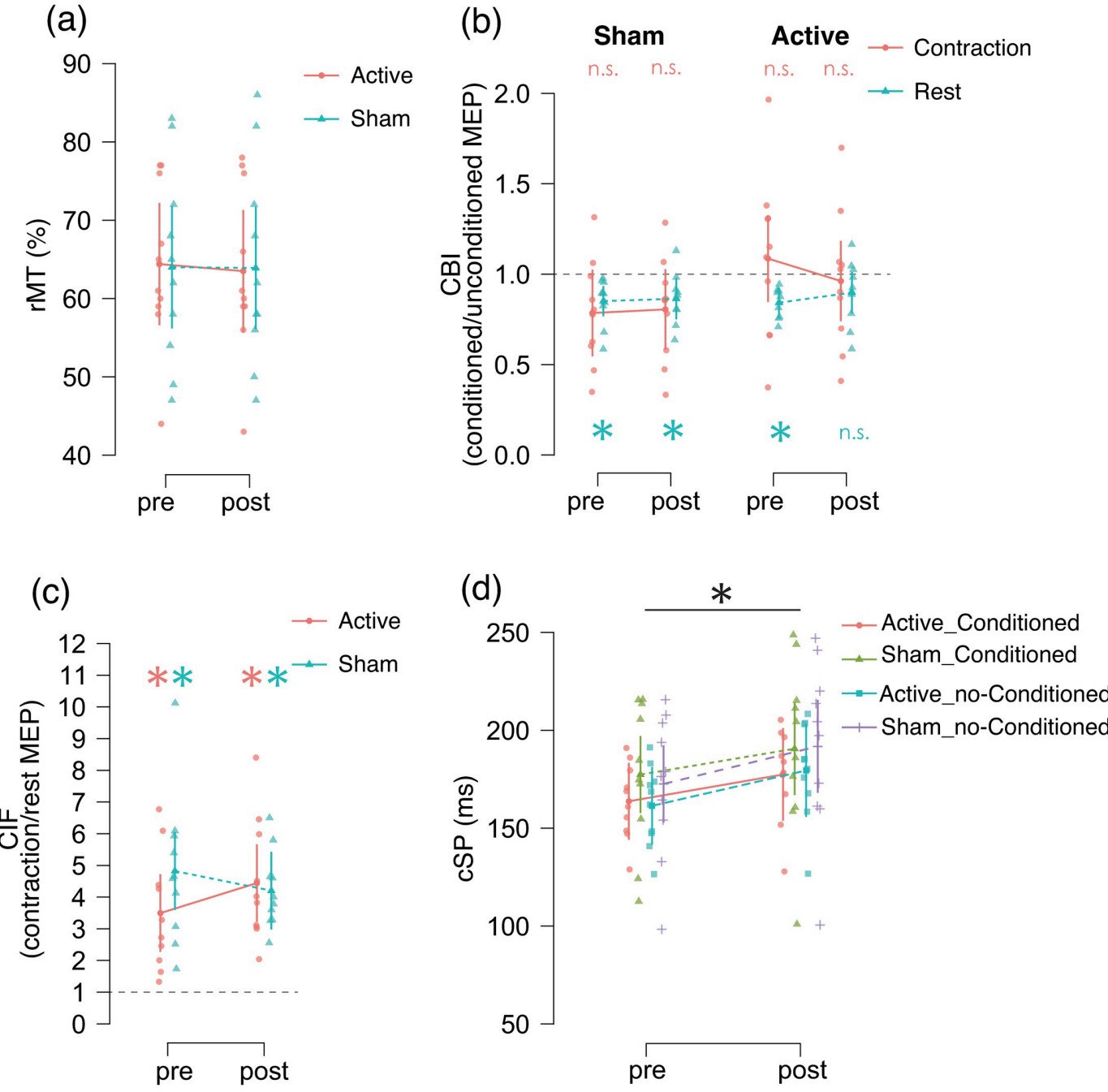

**Fig 2.** (A) Resting motor threshold (rMT), (B) Cerebellar brain inhibition (CBI), (C) Contraction-induced facilitation (CIF), and (D) Cortical silent period (cSP). (A) rMT before and after repetitive transcranial magnetic stimulation (rTMS): The vertical scale indicates rMT (percentage of maximum output: %MO). The red circles and blue triangles indicate participants' mean rMT. Error bars indicate 95% confidence intervals in groups. (B) CBI in the Sham-rTMS and Active-rTMS groups before and after rTMS: The vertical scale indicates CBI (conditioned/unconditioned). The red circles and blue triangles indicate the participant mean CBI under the rest and contraction conditions, and error bars indicate 95% confidence intervals. Asterisks indicate significant inhibition. (C) CIF before and after rTMS: The vertical scale indicates CIF (contraction/rest). The red circles and blue triangles indicate the mean CIF of the participants in the sham rTMS and active rTMS conditions, and error bars indicate 95% confidence intervals. The asterisks indicate significant facilitation. (D) cSP in the Sham-rTMS and Active-rTMS groups under conditioned and non-conditioned states, respectively, by cerebellar single-pulse TMS for cerebellar brain inhibition. Error bars indicate 95% confidence intervals. Asterisk indicates a significant difference between pre- and post-stimulation.

**Table 1. Result of rMT for motor cortex stimulation.**

| | | | | 95% CI | |
|---|---|---|---|---|---|
| Time | rTMS | Mean | SE | Lower | Upper |
| Pre | Active | 64.4 | 3.535 | 57.471 | 71.329 |
| Post | Active | 63.5 | 3.535 | 56.571 | 70.429 |
| Pre | Sham | 64 | 3.535 | 57.071 | 70.929 |
| Post | Sham | 63.9 | 3.535 | 56.971 | 70.829 |

rMT: resting motor threshold; rTMS: repetitive transcranial magnetic stimulation; CI: Confidence interval; SE: standard error

**Table 2. Result of rMT for cerebellar stimulation.**

| | | | | 95% CI | |
|---|---|---|---|---|---|
| Time | rTMS | Mean | SE | Lower | Upper |
| Pre | Active | 34.9 | 1.278 | 31.196 | 38.604 |
| Post | Active | 34.5 | 1.293 | 30.796 | 38.204 |
| Pre | Sham | 36.3 | 2.226 | 32.596 | 40.004 |
| Post | Sham | 36.2 | 2.255 | 32.496 | 39.904 |

rMT: resting motor threshold; rTMS: repetitive transcranial magnetic stimulation; CI: Confidence interval; SE: standard error

likelihood = -77.617, df = 6, AIC = 167.233, and BIC = 177.367. These results indicate that this model is acceptable for testing the effects of rTMS and time and excludes the random effects of the grouping factor: "participant." This model indicated no fixed effect of time (Estimate = -0.082, SE = 0.193, df = 18, t = -0.425, p = 0.676), rTMS (Estimate = -0.272, SE = 0.375, df = 18, t = -0.726, p = 0.477), and interaction between time and rTMS (Estimate = -0.392, SE = 0.193, df = 20, t = -2.024, p = 0.058).

## (d) cSP

Fig 2D and Table 6 show the cSP results. The model fit statistics were as follows: deviance = 609.333, log of likelihood = -304.666, df = 15, AIC = 639.333, and BIC = 675.063. These results indicate that this model is acceptable for testing the effects of rTMS, time, and conditioning TMS for CBI, and it excludes the random effects of the grouping factor: "participant." This model indicated a significant fixed effect of time and interaction between time and TMS conditioning (Table 5). Therefore, we conducted post hoc comparison (t-test) because assumption test (Levene's test) indicates equality of variances (F = 0.171, p = 0.915). There was a significant difference in cSP between pre- and post-rTMS (mean difference = -16.032, standard error = 7.276, t = -2.203, p = 0.031). In contrast, no such difference was observed between conditioned and no-conditioned (mean difference = 0.961, standard error = 7.276, t = 0.132, p = 0.895).

## Discussion

In this study, we investigated the effects of cerebellar rTMS on CIF in healthy individuals. Our results indicated that a single session of 1Hz rTMS over the cerebellum immediately reduced CBI in the resting state but had no significant effect on CIF, rMT for motor cortex stimulation,

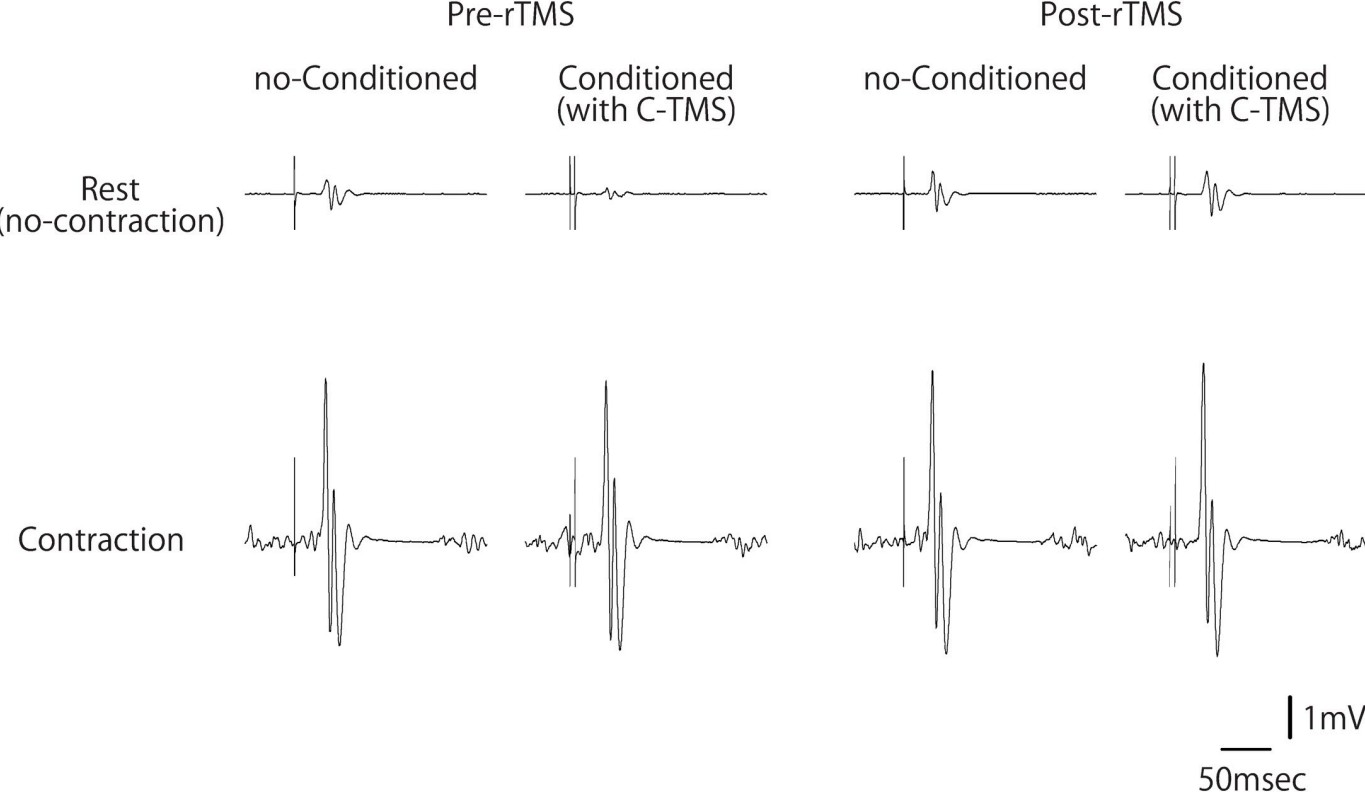

**Fig 3. Typical waveform of motor evoked potentials in one participant in active-repetitive transcranial magnetic stimulation (rTMS) condition.** Upper lines indicate in rest conditions, and lower lines indicate in contraction conditions. The left and right sides indicate pre- and post-rTMS conditions, respectively. The respective left lines show MEPs elicited by test stimulation of the motor cortex alone (no-Conditioned), whereas the respective right lines show MEPs elicited by single-pulse cerebellar TMS (C-TMS) preceding the motor cortex stimulation (conditioned).

rMT for cerebellar stimulation, cSP, and CBI during muscle contraction. These results suggest that CIF function in the primary motor cortex may not be affected by the contralateral cerebellum. 1Hz rTMS over the cerebellum can reduce the cerebellar inhibitory output only in the resting state; however, it cannot modulate the contralateral motor cortex function for

**Table 3. One-sample t-test of CBI.**

| Time | rTMS | Contraction | Mean | SE | 95% CI Lower | 95% CI Upper | z | p† | |
|------|------|-------------|------|-----|-------|-------|---|----|---|
| Pre | Active | Rest | 0.842 | 0.041 | 0.763 | 0.922 | -3.882 | < .001 | * |
| Pre | Sham | Rest | 0.851 | 0.041 | 0.772 | 0.931 | -3.665 | < .001 | * |
| Pre | Active | Contraction | 1.086 | 0.114 | 0.862 | 1.311 | 0.755 | 0.45 | |
| Pre | Sham | Contraction | 0.785 | 0.114 | 0.561 | 1.01 | -1.875 | 0.061 | |
| Post | Active | Rest | 0.898 | 0.055 | 0.79 | 1.006 | -1.846 | 0.065 | |
| Post | Sham | Rest | 0.864 | 0.055 | 0.757 | 0.972 | -2.461 | 0.014 | * |
| Post | Active | Contraction | 0.962 | 0.106 | 0.754 | 1.17 | -0.356 | 0.722 | |
| Post | Sham | Contraction | 0.805 | 0.106 | 0.597 | 1.014 | -1.833 | 0.067 | |

† P-values correspond to the test of null hypothesis against 1.

* Statistically significant.

CBI: cerebellar brain inhibition; rTMS: repetitive transcranial magnetic stimulation; CI: Confidence interval; SE: standard error

**Table 4. Linear mixed-effect model of CBI.**

| Term | Estimate | SE | df | t | p | |
|---|---|---|---|---|---|---|
| Intercept | 0.887 | 0.041 | 18.039 | 21.698 | < .001 | * |
| time | 0.004 | 0.02 | 18.787 | 0.217 | 0.831 | |
| rTMS | 0.06 | 0.041 | 18.039 | 1.477 | 0.157 | |
| rest/cont | 0.023 | 0.037 | 18.086 | 0.62 | 0.543 | |
| time ✻ rTMS | 0.013 | 0.02 | 18.787 | 0.626 | 0.539 | |
| time ✻ rest/cont | 0.022 | 0.014 | 36 | 1.568 | 0.126 | |
| rTMS ✻ rest/cont | 0.054 | 0.037 | 18.086 | 1.467 | 0.16 | |
| time ✻ rTMS ✻ rest/cont | 0.023 | 0.014 | 36 | 1.688 | 0.1 | |

Note: The intercept corresponds to the (unweighted) grand mean; for each factor with k levels, k—1 parameters are estimated with sum contrast coding. Consequently, the estimates cannot be directly mapped to factor levels. We use estimated marginal means to obtain estimates for each factor level/design cell or their differences. CBI: cerebellar brain inhibition; rTMS: repetitive transcranial magnetic stimulation; SE: standard error; df: degrees of freedom.

* Statistically significant

**Table 5. One-sample t-test of CIF.**

| time | rTMS | Estimate | SE | 95% CI Lower | 95% CI Upper | z | p† | |
|---|---|---|---|---|---|---|---|---|
| pre | Active | 3.494 | 0.596 | 2.326 | 4.663 | 4.183 | < .001 | * |
| Post | Active | 4.442 | 0.596 | 3.273 | 5.611 | 5.773 | < .001 | * |
| Pre | Sham | 4.821 | 0.596 | 3.653 | 5.99 | 6.409 | < .001 | * |
| Post | Sham | 4.203 | 0.596 | 3.034 | 5.371 | 5.371 | < .001 | * |

† P-values correspond to the test of null hypothesis against 1.

* Statistically significant.

rTMS: repetitive transcranial magnetic stimulation; CI: Confidence interval; SE: standard error; CIF: contraction-induced facilitation

**Table 6. Linear mixed-effect model of cSP.**

| Term | Estimate | SE | df | t | p | |
|---|---|---|---|---|---|---|
| Intercept | 176.81 | 6.594 | 17.997 | 26.815 | < .001 | * |
| Time | -8.016 | 3.118 | 18.005 | -2.571 | 0.019 | * |
| rTMS | -6.287 | 6.594 | 17.997 | -0.954 | 0.353 | |
| ConditioningTMS | 0.481 | 0.568 | 17.999 | 0.846 | 0.409 | |
| Time ✻ rTMS | 0.079 | 3.118 | 18.005 | 0.025 | 0.98 | |
| Time ✻ ConditioningTMS | 1.298 | 0.4 | 18 | 3.244 | 0.005 | * |
| rTMS ✻ ConditioningTMS | -0.409 | 0.568 | 17.999 | -0.72 | 0.481 | |
| Time ✻ rTMS ✻ ConditioningTMS | -0.197 | 0.4 | 18 | -0.491 | 0.629 | |

Note: The intercept corresponds to the (unweighted) grand mean; for each factor with k levels, k—1 parameters are estimated with sum contrast coding. Consequently, the estimates cannot be directly mapped to factor levels. We use estimated marginal means to obtain estimates for each factor level/design cell or their differences. cSP: cortical silent period; rTMS: repetitive transcranial magnetic stimulation; SE: standard error; df: degrees of freedom.

* Statistically significant.

corticospinal excitability during muscle contraction. The interpretation of these results is not straightforward and should be approached with caution, considering multiple concerns.

Regarding CBI, the linear mixed-effects model revealed no significant marginal or interactive effect of contraction and rTMS on CBI. A one-sample t-test revealed that CBI during rest was found in all conditions without post-active-rTMS, indicating that the MEP at rest was significantly reduced by conditioning single-pulse cerebellar TMS, but this reduction was disrupted by rTMS. Concerning CBI, single-pulse TMS induces action potentials in the Purkinje fiber, leading to inhibition of the dentate nuclei [10]. Deactivation of the dentate-thalamus cortical pathway induces inhibition of excitability in the contralateral corticospinal pathway [10]. The absence of CBI at rest indicates the inhibition of cerebellar output [10, 37]. Therefore, CBI at rest was absent in post-active-rTMS, suggesting that cerebellar rTMS reduces inhibitory output from the cerebellum. This finding is consistent with a previous study [41] and supports the notion that rTMS was accurately targeted to the cerebellum. rTMS may frequently induce action potentials in certain cerebellar neurons, such as parallel fibers, Purkinje neurons, and mossy and climbing fibers, leading to changes in synaptic transmission efficiency similar to that in long-term depression [37]. However, caution is necessary when interpreting this result because it is difficult to identify the exact changes in synaptic transmission efficiency that lead to the observed output change.

CBI was absent during muscle contraction and in both stimulation conditions (sham- and active-rTMS) and timing (pre- and post-rTMS). First, CBI is reportedly decreased during contraction [11, 12]. While the physiological mechanism has not been clarified, some possible hypotheses have been discussed. In the context of active muscles, the cerebellum may diminish the suppression within the motor cortex via the dentate-thalamus cortical pathway, thereby increasing the excitability of the motor cortex and reducing its susceptibility to inhibition by other neural circuits [11, 12]. Furthermore, brain regions that block inhibitory input from the cerebellum may be located within the motor cortex [12]. Based on the se possible mechanism, CBI may be reduced during muscle contraction. In our study, interestingly, CBI during muscle contraction post-active-rTMS did not change, although cerebellar output during rest was reduced by rTMS. These findings suggest the presence of an inhibitory mechanism, possibly outside the cerebellum, that limits cerebellar inhibitory output to enhance corticospinal excitability during muscle contraction. Additionally, long-interval intracortical ICF is known to be reduced in patients with degenerative cerebellar ataxia [49]. While the ICF circuit is influenced by muscle contraction [47], it is not directly affected by conditioning single-pulse cerebellar TMS [3]. Based on these findings, we hypothesized that there is no direct input from the cerebellum to the ICF-related neural circuits that contribute to increased corticospinal tract excitability during muscle contraction, and that input from the cerebellum to neural circuits that increase corticospinal excitability may be gated.

The CIF reflects the corticospinal excitatory function in muscle contraction. The linear mixed-effects model revealed that there was no significant marginal effect of time or rTMS on CIF. The one-sample t-test revealed that the CIF was significantly increased in all conditions. These results indicate that corticospinal excitability is significantly facilitated by voluntary muscle contraction, but this facilitation is not modulated by cerebellar rTMS and/or time. Furthermore, linear mixed-effects models found that rMT and cSP were not significantly changed by rTMS, where cSP reflects the excitability of the corticospinal tract at rest [50] and GABAergic inhibitory neural circuits in the primary motor cortex [16]. rTMS had no effect on any of these indicators, indicating that it is difficult to induce sustained plastic changes in brain regions away from the cerebellum. Although prior studies have reported changes in ICF [37], it may not be possible to alter the ability to manipulate excitability during voluntary contractions.

Nevertheless, we have concerns regarding the results of the CIF in terms of statistical significance. Our result indicated a marginal interaction between time and rTMS on CIF (p = 0.058; Fig 3C). While this interaction did not reach conventional levels of statistical significance, it suggests a remaining potential influence of the cerebellum on CIF. This finding may align with the hypothesis that the cerebellum may play a role in modulating corticospinal excitability during muscle contraction, albeit to a limited extent. Further research with larger sample sizes and varying rTMS parameters could elucidate this interaction more clearly. We have also employed a one-sample t-test to assess the facilitation of corticospinal excitability (CIF) during muscle contraction. This approach, however, may not fully capture the complexity of the null hypothesis, which could encompass both inhibitory ($<1$) and facilitatory ($>1$) responses, as well as changes in magnitude post-rTMS. Our findings may have shown a lack of significant facilitation change; however, it is crucial to note that the one-sample t-test may not adequately reflect the full spectrum of potential physiological effects.

Regarding the cSP, although we could not find the effect of rTMS, there was significant interaction between time and conditioning of cerebellar TMS for CBI. The post-hoc comparison indicated no significant difference in cSP between conditioning and no-conditioning, but there was significant difference between pre- and post-rTMS. These findings suggested that cSP was significantly prolonged over time despite the lack of an effect of rTMS and conditioning single-pulse cerebellar TMS for CBI. cSP can affect the intensity of TMS, but there was no significant change in rMT, indicating that the intensity might not have affected cSP. A previous study reported that cSP could affect mental fatigue [51], and in this study, the force control task was repeated, which may have induced mental fatigue. Therefore, mental fatigue may have affected cSP results.

This study has several limitations. It is reasonable to assume that if there is no difference between groups in linear mixed model analysis, there should be the same suppression in one-sample t-test as in the pre-rTMS condition, but this was not the case in our results. In general, when examining the effects of multiple factors using analysis of variance (ANOVA) or linear mixed models, significant differences are less likely to emerge because the number of groups compared increases. In contrast, simple two-group comparisons have a relatively lower risk rate. The differences in the statistical methods used in the linear mixed model analysis and one-sample t-test, along with the difference in statistical power in this study, may have led to seemingly inconsistent results. Additionally, the intensity of cerebellar TMS for CBI and rTMS over the cerebellum was set to 90% of the rMT in the right FDI relative to the cerebellum [13]. Our result of rMT for cerebellar stimulation was approximately 35% of the maximum output of magnetic stimulator, and the maximum electrical field by cerebellar TMS was 1.23 V/m on cerebellar cortex. Therefore, the magnitude of electrical field may be estimated approximately 0.387 (= 1.23×0.35×0.9) V/m. However, other previous studies have selected even stronger stimulus intensities, i.e., 95% [10] and 120% rMT [22, 43] for cerebellar stimulation. The intensity of rTMS to the cerebellum might be a possible factor in the effect in this examination.

Our findings show that 1Hz cerebellar rTMS can downregulate cerebellar output excitability without affecting the contralateral motor cortex. In individuals with SCD, the function of the contralateral motor cortex and inhibitory output of the cerebellum are changed [6]; therefore, therapeutic targets should consider not only the cerebellum but also the motor cortex. If we also need to modulate motor cortex function in individuals with SCD, we need to consider delivering NIBS to the motor cortex as well because cerebellar rTMS does not modulate contralateral motor cortex function. Our results also support the hypothesis that cerebellar rTMS in patients with SCD reduces ataxia by modulating the excitability of cerebellar cortical and deep nucleus outputs rather than by affecting brain sites other than the cerebellum.

Despite affecting CBI at rest, cerebellar rTMS does not alter the CIF of contralateral corticospinal excitability, rMT, or cSP. The cerebellum's contribution to increased corticospinal excitability during muscle contraction in the contralateral motor cortex may be minimal. However, from a different perspective, rTMS over the cerebellum can produce effects confined to the cerebellum without affecting remote brain regions. The findings provided in this study may guide decisions on the treatment site for cerebellar ataxia.

## Acknowledgments

We thank Editage (www.editage.jp) for English language editing.

## Author Contributions

**Conceptualization:** Akiyoshi Matsugi, Nobuhiko Mori.

**Data curation:** Akiyoshi Matsugi, Aki Tsuzaki, Soichi Jinai.

**Formal analysis:** Akiyoshi Matsugi.

**Funding acquisition:** Akiyoshi Matsugi, Yohei Okada, Nobuhiko Mori.

**Investigation:** Akiyoshi Matsugi, Aki Tsuzaki, Soichi Jinai.

**Methodology:** Akiyoshi Matsugi, Koichi Hosomi.

**Project administration:** Akiyoshi Matsugi.

**Resources:** Akiyoshi Matsugi, Yohei Okada, Nobuhiko Mori, Koichi Hosomi.

**Software:** Akiyoshi Matsugi, Yohei Okada, Nobuhiko Mori, Koichi Hosomi.

**Supervision:** Yohei Okada, Nobuhiko Mori, Koichi Hosomi.

**Validation:** Akiyoshi Matsugi, Nobuhiko Mori.

**Visualization:** Akiyoshi Matsugi.

**Writing – original draft:** Akiyoshi Matsugi.

**Writing – review & editing:** Akiyoshi Matsugi, Aki Tsuzaki, Soichi Jinai, Yohei Okada, Nobuhiko Mori, Koichi Hosomi.

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
