## [Decision Letter · Decision Letter 0]

14 May 2024

PONE-D-23-37474No effect of cerebellar repetitive transcranial magnetic stimulation on contraction-induced facilitation of corticospinal excitabilityPLOS ONE

Dear Dr. Matsugi,

Thank you for submitting your manuscript to PLOS ONE. After careful consideration, we feel that it has merit but does not fully meet PLOS ONE’s publication criteria as it currently stands. Therefore, we invite you to submit a revised version of the manuscript that addresses the points raised during the review process.

Please make sure to address the concerns of the reviewers: primarly considering methodology/design clarifications (such as inclusion subjects with no CBI), statistical analysis (please see comments of the reviewer 1 ), and the language use. 

We look forward to receiving your revised manuscript.

Kind regards,

Maria Nazarova

Academic Editor

PLOS ONE

Journal Requirements:

 [JSPS KAKENHI (grant number: 23K10418)].  

3. We note that Figure 1 in your submission contain copyrighted images. All PLOS content is published under the Creative Commons Attribution License (CC BY 4.0), which means that the manuscript, images, and Supporting Information files will be freely available online, and any third party is permitted to access, download, copy, distribute, and use these materials in any way, even commercially, with proper attribution. For more information, see our copyright guidelines: http://journals.plos.org/plosone/s/licenses-and-copyright.

Reviewers' comments:

Reviewer's Responses to Questions

**Comments to the Author**

1. Is the manuscript technically sound, and do the data support the conclusions?

Reviewer #1: Partly

Reviewer #2: Yes

2. Has the statistical analysis been performed appropriately and rigorously? 

Reviewer #1: No

Reviewer #2: Yes

3. Have the authors made all data underlying the findings in their manuscript fully available?

Reviewer #1: Yes

Reviewer #2: Yes

4. Is the manuscript presented in an intelligible fashion and written in standard English?

Reviewer #1: Yes

Reviewer #2: Yes

5. Review Comments to the Author

Reviewer #1: The authors investigated the effects of cerebellar rTMS on contraction-induced facilitation (CIF), defined as the ratio of MEP amplitude during contraction to that during rest. The results predominantly yielded negative outcomes, indicating no significant effect of cerebellar rTMS on CIF, rMT, sCP, and CBI. Several major concerns arise from this study.

While the introduction effectively outlines the general role of the cerebellum, the rationale for investigating CIF of corticospinal excitability within the context of this cerebellar study remains unclear. CIF represents a unique condition wherein subjects are required to sustain muscle contraction, which may not reflect a natural physiological state. Given this, it is essential for the authors to clarify why CIF was chosen as a focal point in this cerebellar study and elucidate its significance. Specifically, understanding the motivation behind investigating CIF and its relevance to cerebellar function and clinical implications would strengthen the rationale for the study design. Additionally, providing a thorough explanation of the physiological background underlying CIF and its implications for cerebellar modulation of corticospinal excitability would enhance the interpretation of the study's findings.

Some subjects exhibited CBI ratios exceeding 1 during pre-rTMS contraction (Fig.2b, red circles). CBI is defined as MEP inhibition (<1) in response to conditioned stimuli relative to MEP amplitude during test stimuli. It is imperative for the authors to elucidate the reasons behind this occurrence and explain why these subjects were not excluded, especially in light of the criteria used to exclude two subjects due to resting CBI. If this occurrence is validated, I would inquire about the physiological implications of such a finding in relation to the study outcomes.

Of greater concern is the statistical disparity between the one-sample test and linear mixed model analyses. The former suggested that rTMS reduced CBI during the resting state, while no significant effects on CIF, rMT, cSP, and CBI during muscle contraction were observed (The null hypothesis for CBI is equalto 1, indicating no inhibition; however, it is important to clarify whether this is tested using one-sided or two-sided). Conversely, the latter statistical approach did not corroborate these findings, indicating no effects of time, TMS intervention, or contraction on CBI. Given these inconsistent results, the authors arbitrarily concluded a positive effect of rTMS on the cerebellum at rest, and a negative effect on motor function during muscle contraction. However, the statistical validity of these conclusions is debatable. In addition, it is imperative for the authors to carefully elaborate on the interpretation of negative findings, particularly the absence of rTMS effects on the cerebellum during muscle contraction. Several potential pitfalls, including technical issues related to data acquisition (coil parameters) and statistical power, should be thoroughly addressed.

Reviewer #2: In this manuscript, the authors tested the involvement of the cerebellum in the contraction-induced facilitation (CIF) with TMS and 1Hz rTMS. They probed the CIF, CBI, CSP, and RMT before and after rTMS of the right CB. They report no change in any of the parameters except CBI at rest, which was reduced.

As a general impression, the manuscript could benefit from a (complete) rewriting by a native speaker, a professional writer, or an AI tool (just one example among many, L.45-47 "...evidence of the effects of cerebellar degeneration and modulation of cerebellar output excitability in the motor cortex or corticospinal excitability during force control remains lacking." is intelligible but quite awkward to read. This observation unfortunately applies to almost every single phrase in the manuscript)

The study is well-designed and conducted, and the conclusions drawn are sound. The observation that 1Hz rTMS over the cerebellum can reduce cerebellar inhibitory output towards M1, yet does not modulate the excitability of the contralateral M1, is particularly relevant given that this effect remains debated. However, a major question arises concerning the manuscript's suggestion to use NIBS (specifically rTMS) for modulating cerebellar output in SCD patients: what are the bases for expecting that the cerebellar circuits in SCD patients might respond to NIBS in a manner similar to those of healthy individuals? This is suggested towards the end of the discussion (L349-359).

The authors might also want to address the following minor points:

1. An increased RMT does not necessarily imply suppression of corticospinal excitability.

2. Please avoid repeating information from the main text in the figure legends; choose one location for each piece of information.

3. Use the explicit description "1Hz rTMS" instead of the more general "low-frequency" rTMS

6. PLOS authors have the option to publish the peer review history of their article (what does this mean?). If published, this will include your full peer review and any attached files.

Reviewer #1: **Yes: **Teppei Matsubara

Reviewer #2: No

---

## [Author Response · Author response to Decision Letter 0]

22 May 2024

Response to Editors’ and reviewers’ comments

We greatly appreciate the helpful suggestions concerning our manuscript titled “No effect of cerebellar repetitive transcranial magnetic stimulation on contraction-induced facilitation of corticospinal excitability”. We have addressed all the concerns of the editor and reviewer, and have carefully revised our manuscript accordingly. The changes are marked using track changes. Below are our point-by-point responses to the reviewers’ comments. Please let us know if further revisions are needed; we would be glad to incorporate them.

Summary of Revisions

Reviewer 1

(Comment #1)

While the introduction effectively outlines the general role of the cerebellum, the rationale for investigating CIF of corticospinal excitability within the context of this cerebellar study remains unclear. CIF represents a unique condition wherein subjects are required to sustain muscle contraction, which may not reflect a natural physiological state. Given this, it is essential for the authors to clarify why CIF was chosen as a focal point in this cerebellar study and elucidate its significance. Specifically, understanding the motivation behind investigating CIF and its relevance to cerebellar function and clinical implications would strengthen the rationale for the study design. Additionally, providing a thorough explanation of the physiological background underlying CIF and its implications for cerebellar modulation of corticospinal excitability would enhance the interpretation of the study's findings.

(Reply to Comment #1)

Thank you very much for your valuable feedback. We understand the need to clarify the rationale for investigating CIF in the context of our current study, and have majorly revised the introduction section.

(Comment #2)

Some subjects exhibited CBI ratios exceeding 1 during pre-rTMS contraction (Fig.2b, red circles). CBI is defined as MEP inhibition (<1) in response to conditioned stimuli relative to MEP amplitude during test stimuli. It is imperative for the authors to elucidate the reasons behind this occurrence and explain why these subjects were not excluded, especially in light of the criteria used to exclude two subjects due to resting CBI. If this occurrence is validated, I would inquire about the physiological implications of such a finding in relation to the study outcomes.

(Reply to Comment #2)

Thank you for your comment. The inhibition of MEP by conditioning stimulation of the cerebellum at rest was certainly confirmed before the experiment, however, CBI at contraction condition was not confirmed before the experiment. Figure 2b shows there were no participants in the absence of CBI at rest in the pre-stimulation condition. This finding consists of inclusion criteria for participants with experiments about CBI at rest. On the other hand, the red circles indicate the contraction conditions, and this condition’s CBI was not checked before the experiment.

(Comment #3)

Of greater concern is the statistical disparity between the one-sample test and linear mixed model analyses. The former suggested that rTMS reduced CBI during the resting state, while no significant effects on CIF, rMT, cSP, and CBI during muscle contraction were observed (The null hypothesis for CBI is equal to 1, indicating no inhibition; however, it is important to clarify whether this is tested using one-sided or two-sided). Conversely, the latter statistical approach did not corroborate these findings, indicating no effects of time, TMS intervention, or contraction on CBI. Given these inconsistent results, the authors arbitrarily concluded a positive effect of rTMS on the cerebellum at rest, and a negative effect on motor function during muscle contraction. However, the statistical validity of these conclusions is debatable. In addition, it is imperative for the authors to carefully elaborate on the interpretation of negative findings, particularly the absence of rTMS effects on the cerebellum during muscle contraction. Several potential pitfalls, including technical issues related to data acquisition (coil parameters) and statistical power, should be thoroughly addressed.

(Reply to Comment #3)

Thank you for your great suggestive advice. As the reviewer said, the interpretation of the statistical results in this study is difficult and we think it also needs to be interpreted with careful considerations. 

Regarding CBI, no significant suppression was observed only in the post-active rTMS condition, which could be due to the effect of rTMS. On the other hand, no significant differences were observed between the groups divided by factors in the liner mixed model analysis, suggesting that rTMS did not have an effect that caused a significant difference between the groups in terms of CBI and CIF. It is very reasonable to assume that if there is no difference between groups in the liner mixed model analysis, there should be the same suppression in the one-sample t-test as in the pre, but this was not the case in our results.

In general, when examining the effects of multiple factors using ANOVA or linear mixed models, significant differences are less likely to emerge as the number of groups compared increases. On the other hand, simple two-group comparisons have a relatively lower risk rate. The differences in the statistical methods used in the linear mixed model analysis and the one-sample t-test in this study may have led to apparently contradictory results.

Linear mixed model analysis was used to test the effect of each factor on CIF and CBI because we wanted to minimise the influence of potentially confounding subjects, but this analysis alone could not test whether a CIF or CBI had been achieved. Therefore, we used one-sample t-test under two-sided was used to test the CIF and CBI were obtained, and we conservatively set the null hypothesis for CIF and CBI are not equal to 1. As both analyses were performed conservatively, we believe that the results themselves are valid. Therefore, we believe that the implications of these results should be interpreted with caution.

We have described the lack of consistency in the results of this statistical approach as a limitation. As the reviewers pointed out, we made sure to include a discussion of statistical power in the first place. We also described the effects of stimulus parameters that may have occurred during data acquisition as a point of concern. 

The effects observed for CBI at rest but not during muscle contraction were discussed on the basis of possible mechanisms in the third paragraph of the Discussion section.

Reviewer 2

(Comment #1)

As a general impression, the manuscript could benefit from a (complete) rewriting by a native speaker, a professional writer, or an AI tool (just one example among many, L.45-47 "...evidence of the effects of cerebellar degeneration and modulation of cerebellar output excitability in the motor cortex or corticospinal excitability during force control remains lacking." is intelligible but quite awkward to read. This observation unfortunately applies to almost every single phrase in the manuscript)

(Reply to Comment #1)

Thank you for your helpful comment. This manuscript was re-checked and re-edited by English language editing service (www.editage.jp). 

(Comment #2)

The study is well-designed and conducted, and the conclusions drawn are sound. The observation that 1Hz rTMS over the cerebellum can reduce cerebellar inhibitory output towards M1, yet does not modulate the excitability of the contralateral M1, is particularly relevant given that this effect remains debated. However, a major question arises concerning the manuscript's suggestion to use NIBS (specifically rTMS) for modulating cerebellar output in SCD patients: what are the bases for expecting that the cerebellar circuits in SCD patients might respond to NIBS in a manner similar to those of healthy individuals? This is suggested towards the end of the discussion (L349-359).

(Reply to Comment #2)

Thank you for your comment. This paragraph was structured in a way that made our explanation very difficult for the reader to understand. We have substantially revised the text of this paragraph to make our explanation easier to understand, based on the reviewer's comment.

(Comment #3)

The authors might also want to address the following minor points:

1. An increased RMT does not necessarily imply suppression of corticospinal excitability.

(Reply to Comment #3)

Thank you for your comment. As the reviewer mentioned, the increase in rMT is not due to a single factor. We have added that there are several possibilities.

(Comment #4)

2. Please avoid repeating information from the main text in the figure legends; choose one location for each piece of information.

(Reply to Comment #4)

Thank you for your suggestion. We have changed the legends to avoid duplication.

(Comment #5)

3. Use the explicit description "1Hz rTMS" instead of the more general "low-frequency" rTMS

(Reply to Comment #5)

Thank you for your suggestion. We have changed from “low-frequency rTMS” to “1 Hz rTMS” in those areas where more specific wording is appropriate.

We have carefully considered your comment and have made major revisions to the manuscript. Thank you again for your valuable and constructive contributions to our research. We are looking forward to hearing from you.

Best regards

---

## [Decision Letter · Decision Letter 1]

23 Jul 2024

PONE-D-23-37474R1No effect of cerebellar repetitive transcranial magnetic stimulation on contraction-induced facilitation of corticospinal excitabilityPLOS ONE

Dear Dr. Matsugi,

Thank you for submitting your manuscript to PLOS ONE. After careful consideration, we feel that it has merit but does not fully meet PLOS ONE’s publication criteria as it currently stands. Therefore, we invite you to submit a revised version of the manuscript that addresses the points raised during the review process.

Please, consider the suggested inquires about the "CBI" facilitation interpretation and the statistical approach. 

We look forward to receiving your revised manuscript.

Kind regards,

Maria Nazarova

Academic Editor

PLOS ONE

Journal Requirements:

Reviewers' comments:

Reviewer's Responses to Questions

**Comments to the Author**

1. If the authors have adequately addressed your comments raised in a previous round of review and you feel that this manuscript is now acceptable for publication, you may indicate that here to bypass the “Comments to the Author” section, enter your conflict of interest statement in the “Confidential to Editor” section, and submit your "Accept" recommendation.

Reviewer #1: (No Response)

Reviewer #2: All comments have been addressed

2. Is the manuscript technically sound, and do the data support the conclusions?

Reviewer #1: Partly

Reviewer #2: Yes

3. Has the statistical analysis been performed appropriately and rigorously? 

Reviewer #1: Yes

Reviewer #2: Yes

4. Have the authors made all data underlying the findings in their manuscript fully available?

Reviewer #1: Yes

Reviewer #2: Yes

5. Is the manuscript presented in an intelligible fashion and written in standard English?

Reviewer #1: Yes

Reviewer #2: Yes

6. Review Comments to the Author

Reviewer #1: The revised paper has significantly improved in explaining the relationship between CIF and cerebellar function. However, several concerns remain.

(1) As previously mentioned, the physiological meaning of CBI exceeding 1 during contraction is unclear. While CBI exhibits inhibitory function at rest, the authors do not explain its physiological significance during contraction. Approving such occurrences in multiple subjects across different conditions (sham/active, pre/post), the statistical analysis was conducted assuming CBI is below 1 on both sides. Please explain why a one-sided t-test is valid in a setting where several subjects with CBI exceeding 1 during contraction are not excluded.

(2) If my understanding is correct, for measuring CBI, the test stimuli were delivered at 90% of the rMT in relation to the primary motor cortex (P11, line 196). Does this provide sufficient MEP amplitude to evaluate CBI accurately? Additionally, what intensity was the conditioning stimulus delivered to stimulate the cerebellum after measuring the rMT for cerebellar TMS? Was there any change in the rMT for the cerebellum after rTMS, considering that 1-Hz rTMS over the cerebellum immediately reduced motor function (P11, line 205)?

I recommend that the authors show illustrative examples of the changes or lack thereof in CBI, CIF, and cSP.

(3) Please provide an explanation of the null hypothesis regarding the one-sample t-test in cSP. The data analysis section does not describe that a one-sample t-test was conducted for cSP, but the results section showed these results. If this is valid, please provide p values in Table 5. Additionally, please add an interpretation in the discussion regarding the findings from the linear mixed model, which showed a significant fixed effect of time and interaction between time and TMS conditioning.

(4) Overall, I would recommend emphasizing the statistical considerations in this paper, particularly the negative results regarding the cerebellar contribution to CIF. I especially consider the marginal results on the interaction between time and rTMS in CIF (p = 0.058) in the linear mixed-effect model. In addition, the Figure 3c implies this interaction. The one-sample t-test for CIF does not make much sense to me because the null hypothesis may include not only two sides (<1 indicating inhibition, >1 indicating facilitation) but also the change of its magnitude after rTMS. The negative finding may not necessarily indicate the lack of physiological effect. I would recommend emphasizing the methodological limitations regarding this once again.

(5) In Figure. 1b, please specify the intensity of cerebellar stimulation in relation to the rMT of the cerebellum used to generate this electrical field, and provide evidence supporting its validity for stimulating/modulating the cerebellum within this electrical field.

Minor comments

P11, line 199: "To measure CBI, the optimal ISI at which maximal MEP inhibition was selected was 4-9 ms." The word "selected" should be replaced with "observed."

P11, line 214: "Finally, rTMS analysis was performed..." The term "analysis" should be replaced with "intervention" or a similar term.

Reviewer #2: The authors addressed all points raised and the manuscript revised appropriately. The findings add an important piece of the puzzle the cerebellum represents for the exact conditions in which its control is exerted during a motor command.

This reviewer does not have further comments.

7. PLOS authors have the option to publish the peer review history of their article (what does this mean?). If published, this will include your full peer review and any attached files.

Reviewer #1: **Yes: **Teppei Matsubara

Reviewer #2: No

---

## [Author Response · Author response to Decision Letter 1]

6 Aug 2024

Response to Editors’ and reviewers’ comments

We greatly appreciate the helpful suggestions concerning our manuscript titled “No effect of cerebellar repetitive transcranial magnetic stimulation on contraction-induced facilitation of corticospinal excitability”. We have addressed all the concerns of the editor and reviewer, and have carefully revised our manuscript accordingly. The changes are marked using track changes. Below are our point-by-point responses to the reviewers’ comments. Please let us know if further revisions are needed; we would be glad to incorporate them.

Summary of Revisions

Reviewer 1

(Comment #1)

(1) As previously mentioned, the physiological meaning of CBI exceeding 1 during contraction is unclear. While CBI exhibits inhibitory function at rest, the authors do not explain its physiological significance during contraction. Approving such occurrences in multiple subjects across different conditions (sham/active, pre/post), the statistical analysis was conducted assuming CBI is below 1 on both sides. Please explain why a one-sided t-test is valid in a setting where several subjects with CBI exceeding 1 during contraction are not excluded.

(Reply to Comment #1)

Thank you very much for your valuable feedback. Previous studies have also reported that CBI does not occur during muscle contraction (Kassavetis et al. 2011, and Kassavetis et al. 2016), but the mechanism is not yet clear. We have added a hypothesis to the Discussion about the possible mechanism, which is also discussed in previous studies (lines 395-403 in revised manuscript with track changes).

Since the exclusion criterion is no-observation of CBI at rest, the exclusion criterion is not met even if CBI does not occur during muscle contraction. Therefore, in the contraction condition, cases exceeding 1 cannot be excluded. Such cases exceeding 1 during contraction are expected to be included from the range of error bars presented in the graphs of previous studies (Kassavetis et al. 2011, and Kassavetis et al. 2016). 

The test we used is a two-sides test for CBI: we are testing the null hypothesis that CBI is not 1. We did not used one-side test of one-sample test. It was already described in the methods (lines 215-216), but for greater clarity, it was also stated in the results that it was a two-tailed test (line 285).

(Comment #2)

(2) If my understanding is correct, for measuring CBI, the test stimuli were delivered at 90% of the rMT in relation to the primary motor cortex (P11, line 196). Does this provide sufficient MEP amplitude to evaluate CBI accurately? Additionally, what intensity was the conditioning stimulus delivered to stimulate the cerebellum after measuring the rMT for cerebellar TMS? Was there any change in the rMT for the cerebellum after rTMS, considering that 1-Hz rTMS over the cerebellum immediately reduced motor function (P11, line 205)?

I recommend that the authors show illustrative examples of the changes or lack thereof in CBI, CIF, and cSP.

(Reply to Comment #2)

Thank you very much for your valuable feedback. This is completely error of our mention. "the left primary motor cortex" should be corrected to "the right cerebellum". To induce test MEP on right FDI muscle, we stimulate the left primary motor cortex, and the stimulation intensity used in this experiment was set to elicit a test MEP size of 0.5–1 mV to measure CBI before experiment. This explanation had been described in previous section (Single-pulse TMS over the primary motor cortex (test stimulation)) (lines 171-174).

For cerebellar stimulation, the intensity was set to 90% of rMT with double cone coil on right cerebellum. The rMT for cerebellar TMS was defined as the lowest stimulation intensity producing a short-latency EMG response and cervicomedullary MEP in the right FDI muscle immediately after cerebellar TMS for five of ten consecutive stimuli (Tanaka et al. 2017). This explanation had been described in same section (Single-pulse TMS over the cerebellum (conditioning stimulation)) (lines 187-192).

The rMT for the cerebellum was not significantly changed after rTMS. This result was described in Result section ((a) rMT) (lines 237-242).

We added the typical waveform of MEP as Fig3. 

We believe that your advice has made the results of this study more understandable to our readers. Thank you once again for your suggestable advice.

(Comment #3)

(3) Please provide an explanation of the null hypothesis regarding the one-sample t-test in cSP. The data analysis section does not describe that a one-sample t-test was conducted for cSP, but the results section showed these results. If this is valid, please provide p values in Table 5. Additionally, please add an interpretation in the discussion regarding the findings from the linear mixed model, which showed a significant fixed effect of time and interaction between time and TMS conditioning.

(Reply to Comment #3)

We are very sorry; we did not apply the one-sample t-test to the cSP; the table of averages etc. output mechanically by JASP was included as is. This is what is described in Fig. 2(d), so table5 has been deleted.

We did not attend to the importance of this interaction until the reviewers pointed it out to us. To further clarify the effect of this interaction, we conducted a post-hoc comparison. The results showed no effect of conditioning stimulation, only a clear effect of time. In the previous version of the manuscript, the discussion of the cSP results was only included as the limitation, but considering the results of this additional analysis, a new paragraph has been added to the discussion (lines 436-443). In adding the statistical analysis, the methods (lines 220-221) and results (lines 343-347) have also been changed. Again, thanks to the reviewers' remarks, our errors were corrected, and we were able to add additional analyses that provide further evidence. We would like to thank them once again.

(Comment #4)

(4) Overall, I would recommend emphasizing the statistical considerations in this paper, particularly the negative results regarding the cerebellar contribution to CIF. I especially consider the marginal results on the interaction between time and rTMS in CIF (p = 0.058) in the linear mixed-effect model. In addition, the Figure 3c implies this interaction. The one-sample t-test for CIF does not make much sense to me because the null hypothesis may include not only two sides (<1 indicating inhibition, >1 indicating facilitation) but also the change of its magnitude after rTMS. The negative finding may not necessarily indicate the lack of physiological effect. I would recommend emphasizing the methodological limitations regarding this once again.

(Reply to Comment #1)

Thank you very much for your valuable feedback, and we agree this idea. We majorly revised discussion section with adding one paragraph for explanation on these concerns (lines 425-435).

(Comment #5)

(5) In Figure. 1b, please specify the intensity of cerebellar stimulation in relation to the rMT of the cerebellum used to generate this electrical field, and provide evidence supporting its validity for stimulating/modulating the cerebellum within this electrical field.

(Reply to Comment #5)

Thank you for your comment. The rMT for cervicomedullary MEP in Sham and Active-rTMS group at pre- and post-rTMS were 36.3 ± 7.04 % (mean ± standard deviation), 34.9 ± 4.04 %, 36.2 ± 7.13 %, and 34.5 ± 4.09 %, respectively. The electrical field of Fig1(b) was generated by SimNIBS software in the case of using maximum intensity using D-B80 coil connected to magnetic stimulator (MagVenture). The mean of rMT was about 35% of maximum output, indicating about 1.23*0.35*0.9=0.387 V/m may be delivered to cerebellar cortex. The data used for this calculation are described in Methods (line 218), Results (lines 237-242) and further discussion in paragraph of limitation (lines 451-455).

(Comment #6)

Minor comments

P11, line 199: "To measure CBI, the optimal ISI at which maximal MEP inhibition was selected was 4-9 ms." The word "selected" should be replaced with "observed."

P11, line 214: "Finally, rTMS analysis was performed..." The term "analysis" should be replaced with "intervention" or a similar term.

(Reply to Comment #1)

Thank you for your point out our mistake of choice the terms. These terms were revised based on the reviewers’ comments.

We have carefully considered your comment and have made major revisions to the manuscript. Thank you again for your valuable and constructive contributions to our research. We are looking forward to hearing from you.

Best regards

---

## [Decision Letter · Decision Letter 2]

27 Aug 2024

Cerebellar repetitive transcranial magnetic stimulation has no effect on contraction-induced facilitation of corticospinal excitability

PONE-D-23-37474R2

Dear Dr. Matsugi,

We’re pleased to inform you that your manuscript has been judged scientifically suitable for publication and will be formally accepted for publication once it meets all outstanding technical requirements.

Kind regards,

Rita Bella

Academic Editor

PLOS ONE

Additional Editor Comments (optional):

Reviewers' comments:

Reviewer's Responses to Questions

**Comments to the Author**

1. If the authors have adequately addressed your comments raised in a previous round of review and you feel that this manuscript is now acceptable for publication, you may indicate that here to bypass the “Comments to the Author” section, enter your conflict of interest statement in the “Confidential to Editor” section, and submit your "Accept" recommendation.

Reviewer #1: All comments have been addressed

Reviewer #2: All comments have been addressed

2. Is the manuscript technically sound, and do the data support the conclusions?

Reviewer #1: Yes

Reviewer #2: Yes

3. Has the statistical analysis been performed appropriately and rigorously? 

Reviewer #1: Yes

Reviewer #2: Yes

4. Have the authors made all data underlying the findings in their manuscript fully available?

Reviewer #1: No

Reviewer #2: Yes

5. Is the manuscript presented in an intelligible fashion and written in standard English?

Reviewer #1: Yes

Reviewer #2: Yes

6. Review Comments to the Author

Reviewer #1: I am mostly satisfied with the revised manuscript after correcting minor typos.

Please change 'the se' to 'these' on Page 20, Line 341.

Additionally, in Figures 1 and 2, each label should be capitalized (A, B, C, and D) to correspond with the manuscript.

Reviewer #2: The authors have addressed thoroughly all points raised by the reviewers.

This reviewer does not have any further comments.

7. PLOS authors have the option to publish the peer review history of their article (what does this mean?). If published, this will include your full peer review and any attached files.

Reviewer #1: **Yes: **Teppei Matsubara

Reviewer #2: No

---

## [Editor Report · Acceptance letter]

30 Aug 2024

PONE-D-23-37474R2 

PLOS ONE

Dear Dr. Matsugi, 

I'm pleased to inform you that your manuscript has been deemed suitable for publication in PLOS ONE. Congratulations! Your manuscript is now being handed over to our production team.

Kind regards, 

on behalf of

Prof. Rita Bella 

Academic Editor

PLOS ONE